# *Ligilactobacillus salivarius* 7247 Strain: Probiotic Properties and Anti-*Salmonella* Effect with Prebiotics

**DOI:** 10.3390/antibiotics12101535

**Published:** 2023-10-12

**Authors:** Vyacheslav M. Abramov, Igor V. Kosarev, Andrey V. Machulin, Evgenia I. Deryusheva, Tatiana V. Priputnevich, Alexander N. Panin, Irina O. Chikileva, Tatiana N. Abashina, Ashot M. Manoyan, Anna A. Ahmetzyanova, Olga E. Ivanova, Tigran T. Papazyan, Ilia N. Nikonov, Nataliya E. Suzina, Vyacheslav G. Melnikov, Valentin S. Khlebnikov, Vadim K. Sakulin, Vladimir A. Samoilenko, Alexey B. Gordeev, Gennady T. Sukhikh, Vladimir N. Uversky

**Affiliations:** 1Federal Service for Veterinary and Phytosanitary Surveillance (Rosselkhoznadzor) Federal State Budgetary Institution “The Russian State Center for Animal Feed and Drug Standardization and Quality” (FGBU VGNKI), 123022 Moscow, Russia; kosarev-52@mail.ru (I.V.K.);; 2Kulakov National Medical Research Center for Obstetrics, Gynecology and Perinatology, Ministry of Health, 117997 Moscow, Russia; 3Skryabin Institute of Biochemistry and Physiology of Microorganisms, Federal Research Center “Pushchino Scientific Center for Biological Research of Russian Academy of Science”, Russian Academy of Science, 142290 Pushchino, Russia; 4Institute for Biological Instrumentation, Federal Research Center “Pushchino Scientific Center for Biological Research of Russian Academy of Science”, Russian Academy of Science, 142290 Pushchino, Russia; 5Laboratory of Cell Immunity, Blokhin National Research Center of Oncology, Ministry of Health RF, 115478 Moscow, Russia; irinatchikileva@mail.ru; 6Alltech Company, 105062 Moscow, Russia; 7Federal State Educational Institution of Higher Professional Education Moscow State Academy of Veterinary Medicine and Biotechnology Named after K.I. Skryabin, 109472 Moscow, Russia; 8Gabrichevsky Research Institute for Epidemiology and Microbiology, 125212 Moscow, Russia; 9Institute of Immunological Engineering, 142380 Lyubuchany, Russia; 10Department of Molecular Medicine, Morsani College of Medicine, University of South Florida, Tampa, FL 33612, USA; vuversky@usf.edu

**Keywords:** *Ligilactobacillus salivarius*, probiotics, gene expression, bacteriocins, symbiotics, *Salmonella*

## Abstract

The *Ligilactobacillus salivarius* 7247 (LS7247) strain, originally isolated from a healthy woman’s intestines and reproductive system, has been studied for its probiotic potential, particularly against *Salmonella* Enteritidis (SE) and *Salmonella* Typhimurium (ST) as well as its potential use in synbiotics. LS7247 showed high tolerance to gastric and intestinal stress and effectively adhered to human and animal enterocyte monolayers, essential for realizing its probiotic properties. LS7247 showed high anti-*Salmonella* activity. Additionally, the cell-free culture supernatant (CFS) of LS7247 exhibited anti-*Salmonella* activity, with a partial reduction upon neutralization with NaOH (*p* < 0.05), suggesting the presence of anti-*Salmonella* factors such as lactic acid (LA) and bacteriocins. LS7247 produced a high concentration of LA, reaching 124.0 ± 2.5 mM after 48 h of cultivation. Unique gene clusters in the genome of LS7247 contribute to the production of Enterolysin A and metalloendopeptidase. Notably, LS7247 carries a plasmid with a gene cluster identical to human intestinal strain *L. salivarius* UCC118, responsible for class IIb bacteriocin synthesis, and a gene cluster identical to porcine strain *L. salivarius* P1ACE3, responsible for nisin S synthesis. Co-cultivation of LS7247 with SE and ST pathogens reduced their viability by 1.0–1.5 log, attributed to cell wall damage and ATP leakage caused by the CFS. For the first time, the CFS of LS7247 has been shown to inhibit adhesion of SE and ST to human and animal enterocytes (*p* < 0.01). The combination of Actigen prebiotic and the CFS of LS7247 demonstrated a significant combined effect in inhibiting the adhesion of SE and ST to human and animal enterocytes (*p* < 0.001). These findings highlight the potential of using the LS7247 as a preventive strategy and employing probiotics and synbiotics to combat the prevalence of salmonellosis in animals and humans caused by multidrug resistant (MDR) strains of SE and ST pathogens.

## 1. Introduction

*Salmonella* is a Gram-negative, facultative anaerobic, intracellular pathogen that belongs to the Enterobacteriaceae family. Currently, more than 2600 serotypes of *Salmonella* are known [1]. The dominant nontyphoidal serovars of *S. enterica*, Enteritidis (SE), and Typhimurium (ST) are common foodborne pathogens responsible for 93.8 million cases of gastroenteritis, 155,000 deaths annually worldwide, and significant economic losses [2,3,4]. SE and ST are considered socially significant zoo-anthroponotic infections as they can cause gastrointestinal diseases in various host species and severe infections in infants, the elderly, and immunocompromised individuals [5]. These pathogens are primarily associated with the consumption of contaminated poultry meat and raw eggs [6]. In the late 20th century, SE emerged as a major egg-associated pathogen [7]. Epidemiologic data from the United States, European Union, England, Wales, Germany, Canada, and other countries indicate that SE has filled the ecologic niche left by the eradication of the *S. enterica* serovar Gallinarum in poultry, resulting in an epidemic rise in human infections [8,9,10,11,12,13]. Between 1996 and 1999, there was a 44% increase in the number of reported cases of human SE infections associated with food products [14]. SE infects table eggs through horizontal transmission from infected laying hens’ feces, vertical transmission via the yolk, protein, or eggshell membranes before egg laying, and contamination of the eggshell after laying [15,16]. SE has become a significant global concern for food safety [12,17].

The widespread use of antibiotics as feed additives to promote the growth of farm animals has contributed to the emergence of multidrug-resistant (MDR) pathogenic microorganisms, including *Salmonella* [18,19,20,21,22,23,24,25,26]. Antibiotics have traditionally been employed as a strategy to combat *Salmonella* infections. However, the frequent and prolonged use of antibiotics not only increases antibiotic resistance among *Salmonella* serovars but also disrupts the normal intestinal microbiota [27]. Antibiotic-resistant SE and ST strains in farm animals can directly transmit to humans through the food chain, or indirectly transfer their resistance genes to human pathogens using mobile genetic elements associated with conjugative plasmids [28]. The rapid global spread of MDR pathogens poses a significant threat to humans and animals, necessitating the development and implementation of alternative methods to antibiotics for combating these pathogens [29]. One promising alternative approach is the potential preventive and therapeutic use of probiotics, prebiotics, and synbiotics against various enteropathogens, including *Salmonella*.

The anti-*Salmonella* activity of lactic acid bacteria (LAB) and their potential as probiotic feed additives, particularly for combating salmonellosis, have been demonstrated [30,31,32]. The antagonistic activity exhibited by lactobacilli as probiotics is a significant functional characteristic that benefits the microbiome of both humans and animals. The antimicrobial effects of LAB may arise from the action of either a single compound or a combination of compounds [33,34,35,36]. For instance, *Lactiplantibacillus plantarum* 1201 has been shown to inhibit intestinal infection by the *S. enterica* subsp. *enterica* serovar Typhimurium strain ATCC 13311 in mice fed a high-fat diet [37]. *Limosilactobacillus fermentum*, *Lactobacillus delbrueckii*, and *L. gasseri* strains isolated from human infants and yogurt have displayed varying in vitro activity against SE [38]. *Ligilactobacillus salivarius* species have evolutionarily adapted to diverse microecological niches within the host organisms (animals and humans) and play a role in enhancing their resistance to pathogens [30,39,40,41].

The anti-*Salmonella* potential of four lactobacilli (*L. salivarius* CECT5713, *L. gasseri* CECT5714, *L. gasseri* CECT5715, and *L. fermentum* CECT5716), isolated from fresh human breast milk, was evaluated in [41]. The findings indicated that all the strains exhibited anti-*Salmonella* properties. *L. salivarius* CECT5713 exhibited the highest level of antibacterial activity in vitro and demonstrated the most significant protective effect against *Salmonella* serotype Choleraesuis in a murine infection model compared to the other tested lactobacillus strains. *L. salivarius* CECT 5713 was originally isolated simultaneously from breast milk and infant feces of a healthy mother–infant pair. The strain was strongly adhesive to Caco-2 cells, produced lactic acid and hydrogen peroxide, and showed a high survival rate after exposition to conditions simulating those found in the gastrointestinal tract [40]. 

Complete genomic sequencing of *L. salivarius* CECT 5713 detected a megaplasmid pHN3 containing six open reading frames (ORFs) that are closely related, but not identical, to the genes responsible for the biosynthesis of salivaricin ABP-118 [42]. The antibacterial activity of *L. salivarius* CECT5713 against SE and ST pathogens has not been studied. *L. salivarius* derived from breast milk has the potential to contribute to the anti-infective protection of newborns and could be considered a candidate for the development of probiotic and synbiotic products for children and young animals [41]. Certain lactobacillus strains, such as *L. salivarius*, *L. fermentum*, and *L. gasseri*, residing in the mother’s intestines during breastfeeding, utilize the entero-mammary pathway to migrate from the intestine to the mammary glands [40,43]. Breast milk contains oligosaccharides that serve as prebiotics, offering protection against intestinal pathogens for infants [44]. A probiotic strain of lactobacilli that has penetrated through the entero-mammary pathway into breast milk forms, together with milk oligosaccharides, a synbiotic that enters the baby’s gastrointestinal tract together with the mother’s milk and enhances the effectiveness of protection against intestinal pathogens [45].

Various virulence genes, found both in the chromosome and plasmids of *Salmonella*, play a crucial role in the pathogenesis of salmonellosis [46,47,48].

The initial stage of *Salmonella* infection involves the adhesion of the pathogen to human and animal enterocytes. The plasmid-encoded fimbriae gene (*pefA*) is responsible for *Salmonella*’s adhesion to the intestinal epithelial cells of the host [49]. Subsequently, the second stage of infection is invasion, facilitated by *Salmonella*’s outer proteins and the hilA virulence genes, which contribute to the invasion of host epithelial cells [46]. *Salmonella* expresses the FimH adhesin of type 1 pili on its surface, allowing interaction with mannose residues present on human and animal intestinal enterocytes [50]. Prebiotics that contain mannose polymers have the ability to inhibit the adhesion of *Salmonella* to enterocytes [51].

The presence of synergistic effects between prebiotics and probiotics in inhibiting the adhesion of *Salmonella* to enterocytes is highly significant [52]. This is important when creating new synbiotics and synbiotic feed additives for the prevention of salmonellosis in humans and animals. While the concept of synbiotics for salmonellosis prevention is promising, it is worth noting that not all combinations of prebiotics and probiotics demonstrate optimal results [53,54]. This highlights the necessity for further research in creating synbiotics with targeted properties. *L. acidophilus*, *L.* gasseri, and *L. salivarius* within the family Lactobacillaceae have evolutionarily adapted to the microecological niches of humans and farm animals. They take part in ensuring the colonization resistance of the digestive system to intestinal pathogens, including *Salmonella* [30,55,56,57,58].

The antimicrobial activity of probiotics is specific to their species and strains, with notable effectiveness against *Salmonella*. Careful screening is necessary for the selection of new effective probiotic strains with pronounced anti-*Salmonella* properties. In order to enhance the antibacterial effects of probiotics, screening for suitable prebiotics and the development of synbiotics is also important.

Thus, the purpose of this work was to select a Lactobacillus strain from the three studied (LS7247, LA7234, and LG7528) with the maximum tolerance to gastrointestinal stress and strong anti-*Salmonella* activity, study its probiotic properties and genetic control of bacteriocin production, and identify a prebiotic for creating a synbiotic with anti-*Salmonella* activity.

## 2. Results and Discussion

### 2.1. Tolerance of Lactobacillus Strains to Gastric and Intestinal Stresses

The tolerance of probiotic microorganisms to gastric and intestinal stress is a crucial requirement for their proper functioning in the host’s digestive system. This section of the research aimed to select the most resistant strain of lactobacillus to gastric and intestinal stress among the three strains tested. The results of the in vitro assessment of the tolerance of *L. animalis* IIE 7234 (LA7234), *L. salivarius* IIE 7247 (LS7247), and *L. gasseri* IIE 7528 (LG7528) to gastric and intestinal stress are presented in Table 1. LA7234 exhibited acceptable tolerance to gastric stress but showed an unacceptable sensitivity to intestinal stress. LG7528 demonstrated good tolerance to gastric stress but was sensitive to intestinal stress at an unacceptable level. In contrast, LS7247 displayed high tolerance to both gastric and intestinal stress. After 60 min of exposure to gastric juice, the LS7247 strain’s degree of resistance to gastric stress was at very good level, with an RD of 2.0. After 5 h of exposure to intestinal juice, the LS7247 strain’s degree of resistance to intestinal stress was at very good level, with an RD of 4.0. 

The challenging environment in the intestines of humans and animals, particularly bile salts, poses difficulties for probiotic strains [59]. The use of comparative genomic analysis by Pan Q. et al. revealed that the mechanisms underlying the tolerance of *L. salivarius* probiotic strains to bile salts are primarily associated with chaperones, the phosphotransferase system (PTS), and peptidoglycan synthesis [60]. The genome of the LS7247 strain contains genes responsible for these functions (2102-15, Accession CP090411.1). A list of “probiotic marker” genes, including those involved in stress resistance (acid, osmotic, oxidative, and temperature), bile salt hydrolase activity, adhesion capacity, and intestinal persistence, was proposed [61,62,63,64].

### 2.2. Anti-Salmonella Activity of Lactobacillus Strains

The anti-*Salmonella* activity of the *L. animalis* IIE 7234 (LA7234), *L. salivarius* IIE 7247 (LS7247), and *L. gasseri* IIE 7528 (LG7528) strains was determined using the delayed antagonism method. The results, presented in Table 2, indicate that all three strains exhibited antagonistic activity against *Salmonella* serovars SE and ST. However, the anti-*Salmonella* activity of the LA7234 and LG7528 strains was significantly lower compared to the LS7247 strain (*p* < 0.05). The LS7247 strain produced inhibition zones against SE pathogens (*S.* Enteritidis ATCC 13076, *S.* Enteritidis ATCC 4931, *S.* Enteritidis IIE Egg 6215, *S.* Enteritidis IIE Egg 6218, and *S.* Enteritidis IIE Egg 6219 strains) ranging from 16.5 ± 0.7 to 18.9 ± 0.6 mm, and against ST pathogens (*S.* Typhimurium ATCC 700720, *S.* Typhimurium ATCC 14028, *S.* Typhimurium IIE BR 6458, and *S.* Typhimurium IIE BR 6461 strains) ranging from 16.9 ± 0.8 to 18.6 ± 0.4 mm. 

The antimicrobial activity against foodborne pathogens is a crucial property of intestinal probiotics as it contributes to the resistance of the gastric and intestinal microecological niches of the host organism [65]. It is worth noting that the known *L. salivarius* strains generally exhibit low anti-*Salmonella* activity [43,66]. In order to further investigate the nature of the anti-*Salmonella* activity of the LS7234 strain, the antibacterial properties of its CFS were examined in subsequent experiments. Given that the LA7234 and LG7528 strains are sensitive to intestinal stress (Table 1) and display low anti-*Salmonella* activity (Table 2), they were excluded from further experiments. 

### 2.3. Anti-Salmonella Activity of CFS from LS7247

The anti-*Salmonella* activity of CFS from LS7247 strain are shown in Table 3. The CFS demonstrated pronounced anti-*Salmonella* activity. Intact CFS-induced inhibition zones of SE and ST pathogens ranged from 14.2 ± 0.4 to 18.5 ± 0.6 mm and 14.5 ± 0.5 to 19.4 ± 0.6 mm, respectively. However, when the CFS was neutralized with NaOH, there was a significant decrease in the level of anti-*Salmonella* activity (*p* < 0.05). Previous studies [67] have shown that a lactic acid concentration of approximately 5 mM can suppress the growth of *S.* Typhimurium and *E. coli* O157:H7. In the present study, the low pH of the medium was attributed to lactic acid produced by the *L. salivarius* culture. This high concentration of lactic acid and low pH allowed *L. salivarius* to inhibit the growth of *S.* Typhimurium and *E. coli* in poultry feed [67]. The main antimicrobial compound produced by *L. rhamnosus* GG against *S.* Typhimurium was lactic acid [68]. Due to the fact that lactic acid contributes to the antibacterial properties of the LS7247 strain against SE and ST pathogens, we studied the dynamics of the production of this short-chain fatty acid in the following experiments. Neutralizing the lactic acid in CFS from the LS7247 strain using NaOH reduced the level of anti-*Salmonella* activity, although it was not completely eliminated. These findings suggest that the residual antibacterial activity of the LS7247 strain could be attributed to other factors, such as the production of bacteriocins. 

### 2.4. Lactic Acid Production by LS7247 Strain

The results of our research, presented in Table 4, demonstrate the production dynamics of lactic acid by the LS7247 strain. After 4 h of cultivation, the concentration of lactic acid in the culture fluid was found to be 3.5 ± 0.4 mM. After 24 h, the concentration of lactic acid increased to 69.7 ± 0.8 mM, and after 48 h, it further rose to 124.0 ± 2.5 mM. Previously, it was found that lactic acid at physiological concentrations (55–110 mM) mediates a potent 10^6^-fold decrease in the viability of 17 different bacterial vaginosis-associated bacteria [69]. Lactic acid was identified as the main antimicrobial compound against *S.* Typhimurium [68,70], *S.* Enteritidis, *E. coli*, and *Listeria monocytogenes* [71]. The production of lactic acid by LAB creates an unfavorable local microenvironment for pathogens [72]. Lactic acid exerts antimicrobial effects by targeting the bacterial cell wall, cytoplasmic membrane, and specific metabolic functions involved in the replication and protein synthesis of pathogens, ultimately leading to their disruption and death [73,74].

### 2.5. Anti-Salmonella Activity of LS7247 Strain Co-Cultivated with SE and ST Pathogens 

The antagonistic activity of the LS7247 strain co-cultivated with SE pathogens (*S.* Enteritidis ATCC 13076, *S.* Enteritidis ATCC 4931, *S.* Enteritidis IIE Egg 6215, *S.* Enteritidis IIE Egg 6218, and *S.* Enteritidis IIE Egg 6219 strains) and ST pathogens (*S.* Typhimurium ATCC 700720, *S.* Typhimurium ATCC 14028, *S.* Typhimurium IIE BR 6458, and *S.* Typhimurium IIE BR 6461 strains) is shown in Table 5. In the co-culture experiments, SE and ST pathogens were sensitive to the bacteriolytic action of the LS7247 strain. Co-cultivation of SE and ST pathogens with the LS7247 strain for 24 h reduced the CFUs of viable test cultures by 1.0–1.5 logs. Gram-negative microorganisms, including *Salmonella*, have an outer membrane (OM) in their cell wall, which acts as an effective permeability barrier against external agents. In most Gram-negative microorganisms, the OM consists of an asymmetric bilayer of phospholipids and lipopolysaccharides, with the latter exclusively found in the outer leaflet [75].

### 2.6. Bacteriocins Produced by LS7247 Strain

The complete genome sequence of LS7247 (2102-15) is available at NCBI GenBank under the accession numbers CP090411:CP090413. A cluster of genes located in the chromosome is responsible for the production of the phage tail protein Enterolysin A and metalloendopeptidase family protein. Enterolysin A belongs to the class III peptidoglycan-degrading bacteriocins that cleave (1→4)-β-linkages between N-acetylmuramic acid and N-acetyl-D-glucosamine residues in a peptidoglycan [76]. Metalloendopeptidase hydrolyzes short peptides connecting the peptidoglycan layers of the bacterial cell wall [77]. The data obtained (Table 2, Table 3 and Table 5) indicate a high anti-*Salmonella* activity for LS7247. Lactic acid produced by the LS7247 strain increases the permeability of *Salmonella* strains’ outer membrane [74], which is necessary for the realization of the antibacterial activity of Enterolysin and metalloendopeptidase against Gram-negative pathogens [78]. 

Salmonellosis refers to nosocomial infections. This is one of the serious problems of medicine. According to the severity of the clinical course (more than 80%), severe and moderate forms of infection prevail in a newborn and elderly [79]. The genes encoding Enterolysin protein and metalloendopeptidase protein have not been found in the genome of the well-characterized human intestinal isolate *L. salivarius* UCC118 [80] and in the genome of the *L. salivarius* P1CEA3 strain, isolated from the gastrointestinal tract of pigs [81]. The decoded genomes of *L. salivarius* strains isolated from humans and animals do not contain the genes encoding Enterolysin protein and metalloendopeptidase protein [82,83,84,85,86]. In this regard, the LS7247 strain is unique in terms of the content of anti-*Salmonella* factors.

A cluster of genes located in the plasmid pLS2102-15 of the LS7247 strain are responsible for the production of class IIb bacteriocin. This bacteriocin belongs to class IIb and consists of two peptides. Both peptides are initially produced as bacteriocin precursors, with the α chain comprising 64 amino acids and the β chain comprising 68 amino acids. These precursor peptides have N-terminal leader sequences of 19 and 22 amino acids, respectively. The mature peptides, after processing, contain 45 and 46 amino acid residues, respectively (Figure 1A). The estimated molecular weight of the α chain is 4950 Da, and the β chain is 5060 Da. The genes encoding the α chain and β chain are genetically linked and are part of the same operon (2102-15, Accession CP090411.1). These findings align with earlier results [87].

In our studies, we compared the primary structures of class IIb two-component bacteriocins produced by LS7247 and other lactobacilli, including the human intestinal isolate *L. salivarius* UCC118. The amino acid sequences of these bacteriocins were found to be identical (Figure 1B). The UCC118 probiotic strain, which was isolated from the human intestines, has been extensively characterized at the molecular level [80]. The bacteriocin from LS7247 is encoded by a gene on the chromosome (2102-15, Accession CP090411.1), while class IIb bacteriocin from UCC118 is encoded by a gene on a 242 kb megaplasmid [88]. Similarly, in another *L. salivarius* strain, SECT 5713, the class IIb bacteriocin is also located on a megaplasmid [43]. The alpha and beta peptide chains of salivaricin P, a class IIb bacteriocin, produced by *L. salivarius* strains isolated from pig feces and ceca (DPC6005, DPC6027, DPC6189, and M7.2, 7.3), were compared to those produced by *L. salivarius* UCC118, isolated from human intestine [89]. 

The production of two-component bacteriocins by *L. salivarius* strains may provide ecological advantages within intestinal bacterial communities [89,90]. Nissen-Meyer and co-authors discovered that the two peptides of class IIb two-peptide bacteriocins form a spiral–spiral structure that penetrates the membrane of intestinal pathogens, leading to membrane leakage and cell death of the pathogens [91]. The increase in antibacterial activity may also be attributed to the presence of genes responsible for the production of additional bacteriocins in the genome of *L. salivarius* strains. The LS7247 strain harbors a plasmid containing genes responsible for the production of a bacteriocin called nisin (Access: CP0904412.1). 

LS7247 produces nisin as a lantibiotic precursor consisting of 59 amino acid residues. It contains a 25-residue N-terminal leader sequence, and the mature nisin consists of 34 residues. The estimated molecular weight of mature nisin is 4350 Da. The newly identified nisin produced by the LS7247 strain exhibits specific substitutions of individual amino acid residues compared to known variants of this peptide, such as nisin P, nisin U, nisin U2, nisin H, nisin Q, nisin F, nisin A, nisin Z, nisin 03, nisin 01-2, nisin J, subtilin, and kunkecin A. The primary structure of nisin produced by the LS7247 strain is identical to the primary structure of nisin S produced by the *L. salivarius* P1CEA3 strain, which was isolated from the gastrointestinal tract of pigs [81]. Nisin exerts its antimicrobial activity by forming pores in the cytoplasmic membrane and inhibiting cell wall synthesis [92]. Figure 2 illustrates the predicted primary structure of nisin from the LS7247 strain.

Bacteria-derived natural antimicrobial compounds, including bacteriocins and organic acids, have gained significant interest as therapeutic alternatives for both humans and animals. Such artificial mixtures show high antibacterial activity to various pathogens [93]. The LS7247 strain possesses the unique capability of producing a mixture of lactic acid, Enterolysin A, metalloendopeptidase, a bacteriocin (class IIb), and the lantibiotic nisin S. This mixture provides the LS7247 strain with high anti-Salmonella activity.

### 2.7. CFS of LS7247 Strain Induces ATP Leakage from SE and ST Pathogens

The CFS of the LS7247 strain induced cell damage in SE and ST pathogen cells, leading to ATP leakage. The levels of ATP leakage, which serve as indicators of cell injury, were investigated and the results are presented in Table 6. Cultivating SE pathogens (*S.* Enteritidis ATCC 13076, *S.* Enteritidis ATCC 4931, *S.* Enteritidis IIE Egg 6215, *S.* Enteritidis IIE Egg 6218, and *S.* Enteritidis IIE Egg 6219 strains) in the presence of LS7247 CFS for 2.5 h increased the extracellular ATP level from 4.5 ± 0.6 (nm/OD) to 28.5 ± 1.3 (nm/OD) (*p* < 0.05). Similarly, cultivating ST pathogens (*S.* Typhimurium ATCC 700720, *S.* Typhimurium ATCC 14028, *S.* Typhimurium IIE BR 6458, and *S.* Typhimurium IIE BR 6461 strains) with LS7247 CFS for 2.5 h increased the extracellular ATP level from 4.0 ± 0.5 (nm/OD) to 29.5 ± 1.2 (nm/OD) (*p* < 0.05). The natural antibacterial complex produced by the LS7247 strain, consisting of lactic acid, a class IIb bacteriocin, and nisin S, increased the permeability of the OM in SE and ST pathogens [94], leading to ATP leakage.

### 2.8. Adhesion of LS7247 Strain to a Monolayer Formed from Human Caco-2, Porcine IPEC-J2, or Chicken Primary Cecal Enterocytes

The studies of the LS7247 strain’s adhesion to the small intestine enterocytes of humans, pigs, and chickens are shown in Table 7. The LS7247 strain demonstrated efficient adhesion to immortalized Caco-2 human intestinal epithelial cells (AA-100%; AI-38.6 ± 2.5), immortalized porcine IPEC-J2 intestinal epithelial cells (AA-100%; AI-32.4 ± 1.9), and chicken primary cecal epithelial cells (AA-100%; AI-27.5 ± 1.6). These findings indicate that the LS7247 strain exhibits high adhesion to enterocytes regardless of the host species (human, pig, or chicken). The ability of the LS7247 strain to adhere strongly to the intestinal epithelium of humans and animals is essential for its colonization and performance of various probiotic functions in the digestive system [95,96,97].

### 2.9. Total Effect of the Actigen Prebiotic and CFS from LS7247 Strain in Inhibiting the Adhesion of SE and ST Pathogens to Caco-2, IPEC-J2, and Chicken Primary Cecal Enterocytes

#### 2.9.1. Inhibiting Effect on Adhesion to Caco-2 Enterocytes

Intensive adhesion of SE and ST pathogens to human Caco-2 enterocytes was detected (Table 8). The adhesion indices of SE and ST pathogens to Caco-2 enterocytes were 25.5 ± 1.2–28.7 ± 1.3 and 26.5 ± 1.2–29.2 ± 1.4, respectively. The Actigen prebiotic inhibited the adhesion of SE and ST pathogens to Caco-2 enterocytes (*p* < 0.01). The adhesion index of SE and ST pathogens to Caco-2 enterocytes decreased 4.0–4.9 times and 3.1–4.3 times, respectively, in relation to the control. For the first time, the CFS from the LS7247 strain was shown to inhibit adhesion of SE and ST pathogens to Caco-2 enterocytes (*p* < 0.01). The adhesion indices of SE and ST pathogens to Caco-2 enterocytes decreased 2.6–4.9 times and 3.0–3.1 times, respectively, in relation to the control. The combination of the Actigen prebiotic and CFS of LS7247 demonstrated a significant combined effect in inhibiting the adhesion of SE and ST pathogens to Caco-2 cells (*p* < 0.001). The adhesion indices of SE and ST pathogens to Caco-2 enterocytes decreased 34.0–34.2 times and 44.9–50.3 times, respectively, in relation to the control. The CFS of the LS7247 strain after incubation together with proteinase K (∆CFS) lost the ability to inhibit the adhesion of SE and ST pathogens to human Caco-2 enterocytes and did not enhance the anti-adhesive effect of the Actigen prebiotic.

#### 2.9.2. Inhibiting Effect on Adhesion to Porcine IPEC-J2 Enterocytes 

Intensive adhesion of SE and ST pathogens to porcine IPEC-J2 enterocytes was observed (Table 9). The adhesion indices of SE and ST pathogens to IPEC-J2 enterocytes ranged from 25.4 ± 1.1 to 29.4 ± 1.5 and 25.8 ± 1.3 to 28.2 ± 1.6, respectively. The presence of the Actigen prebiotic significantly inhibited the adhesion of SE and ST pathogens to IPEC-J2 enterocytes (*p* < 0.01), resulting in a reduction of the adhesion indices by 4.5–4.8 times and 3.8–5.0 times, respectively, compared to the control. Notably, this study revealed for the first time that the CFS from the LS7247 strain had an inhibitory effect on the adhesion of SE and ST pathogens to IPEC-J2 enterocytes (*p* < 0.01). The adhesion indices of SE and ST pathogens to IPEC-J2 enterocytes decreased 3.1–3.5 times and 3.0–3.2 times, respectively, in relation to the control. The combination of the Actigen prebiotic and CFS of LS7247 demonstrated significant total effects in inhibiting the adhesion of SE and ST pathogens to IPEC-J2 enterocytes (*p* < 0.001). The adhesion indices of SE and ST pathogens to IPEC-J2 enterocytes decreased 52.9–60.0 times and 42.1–57.3 times, respectively, in relation to the control. However, it is worth noting that when the CFS of the LS7247 strain was incubated with proteinase K (∆CFS), it lost its ability to inhibit the adhesion of SE and ST pathogens to porcine IPEC-J2 enterocytes and did not enhance the anti-adhesive effect of the Actigen prebiotic. 

#### 2.9.3. Inhibiting Effect on Adhesion to CPCEs 

Intensive adhesion of SE and ST pathogens to chicken primary cecal enterocytes was identified (Table 10). The adhesion indices of SE and ST pathogens to CPCEs were 19.4 ± 1.5–22.3 ± 1.4 and 20.4 ± 1.3–23.6 ± 1.1, respectively. The Actigen prebiotic effectively suppressed the adhesion of SE and ST pathogens to CPCEs (*p* < 0.01). Consequently, the adhesion indices of SE and ST pathogens to CPCEs were reduced by approximately 4.0–4.8 times and 3.6–4.5 times, respectively, compared to the control. For the first time, the CFS from the LS7247 strain was shown to inhibit adhesion of SE and ST pathogens to CPCEs (*p* < 0.01). The adhesion indices of SE and ST pathogens to CPCEs decreased by approximately 3.0–3.5 times and 2.7–3.6 times, respectively, relative to the control. The combination of the Actigen prebiotic and CFS of LS7247 demonstrated a significant combined effect in inhibiting the adhesion of SE and ST pathogens to CPCEs (*p* < 0.001). This combination resulted in a remarkable reduction in adhesion indices for SE and ST pathogens, decreasing by approximately 38.8–55.7 times and 34.0–39.3 times, respectively, in comparison to the control. However, the CFS of the LS7247 strain after incubation together with proteinase K (∆CFS) lost the ability to inhibit the adhesion of SE and ST pathogens to chicken primary cecal enterocytes and did not enhance the anti-adhesive effect of the Actigen prebiotic. 

Thus, we have discovered the anti-adhesive activity of CFS and the combined effect of the Actigen prebiotic and CFS in inhibiting the adhesion of SE and ST pathogens to human Caco-2 enterocytes, porcine IPEC-J2 enterocytes, and chicken primary cecal enterocytes.

The Actigen prebiotic and CFS from the LS7247 strain have a different structure and differ in their mechanisms of action. It is known that the Actigen prebiotic is a component of the yeast cell wall that contains mannose biopolymers (MOSs) [98]. The monosaccharide mannose acts as a ligand for the FimH domain found in type I fimbria of *Salmonella*. The FimH domain is responsible for recognizing mannose patterns on the surface of host enterocytes and facilitating mannose-dependent adhesion of pathogens, including *Salmonella* [50,99,100,101,102]. MOSs bind to the FimH domain, competing with mannose structures on host enterocytes, thus inhibiting the adhesion of pathogens. In this way, it performs a function that mimics the receptor [103,104]. 

The mechanism of anti-adhesive action of LS7247 CFS has not been studied. We assume that the native supernatant contains a protein that interacts with the active site of the FimH domain and reduces its affinity for mannose patterns on the surface of host enterocytes. The consequence of this interaction is a decrease in the adhesion of SE and ST pathogens to enterocytes. Cultivation of the supernatant in conjunction with proteinase K led to proteolysis of proteins, including a protein capable of interacting with the active site of the FimH domain. Proteolysis led to a loss of the anti-adhesive activity by the supernatant (Table 8, Table 9 and Table 10). Further research is required to clarify the mechanisms of the anti-adhesive action of the supernatant. 

SE and ST pathogens are facultative intracellular bacteria. After adhesion, these pathogens could enter cells through a trigger mechanism mediated by a type three secretion system called T3SS-1 or through OM proteins, Rck and PagN, which have been identified as *Salmonella* invasins [105,106,107,108]. Upon invasion of host cells, *Salmonella* triggers inflammation and disrupts the tight junctions of the bowel [51]. Antibiotics have been widely used in clinical settings for the treatment of salmonellosis. However, antibiotic treatment can lead to the emergence of MDR *Salmonella* strains, which can hinder host immune defenses [109]. Therefore, it is crucial to develop novel and reliable methods for preventing salmonellosis. 

The invasion of SE and ST pathogens can only occur after their adhesion to the surface of human and animal enterocytes. Thus, effectively inhibiting the adhesion of SE and ST pathogens to human and animal enterocytes is of paramount importance for the prevention of salmonellosis. This can be achieved through the use of a synbiotic containing the LS7247 strain and Actigen prebiotic, or a feed additive based on them.

## 3. Materials and Methods

### 3.1. Bacterial Strains and Growth Conditions

A complete list of the bacteria used in this work, including strains from the American Type Culture Collection (ATCC) and their growth conditions, is provided in Table 11.

### 3.2. Intestinal Epithelial Cells and Growth Conditions

#### 3.2.1. Caco-2 Human Intestinal Epithelial Cells

Immortalized Caco-2 human intestinal epithelial cells were suspended in culture medium (DMEM plus 10% fetal calf serum (FCS) and 0.02% penicillin and streptomycin each), and were seeded into 12-well cell culture plates at a density of 5 × 10^5^ cells/mL to form a cell monolayer. The plates were incubated for 48 h at 37 °C under 5% CO_2_.

#### 3.2.2. IPEC-J2 Porcine Intestinal Epithelial Cells

The pig, due to its genetic and physiological similarities to humans, is considered a suitable animal model for studying mucosal physiology, including in vitro experiments [110]. An immortalized epithelial cell line from the porcine intestine, IPEC-J2, a relevant in vitro model system for porcine intestinal pathogen–host cell interactions, was used [111]. IPEC-J2 porcine intestinal epithelial cells were suspended in culture medium (DMEM plus 10% FCS and 0.02% penicillin and streptomycin each), and were seeded into 12-well cell culture plates at a density of 5 × 10^5^ cells/mL to form a cell monolayer. The plates were incubated for 48 h at 37 °C under 5% CO_2_.

#### 3.2.3. Chicken Primary Cecal Enterocytes (CPCEs)

Primary epithelial cells were obtained from the ceca of 2-week-old chicks (Kuchinskaya jubilee breed) following an established protocol [112]. The cells were suspended in culture medium (DMEM plus 2.5% fetal calf serum, 0.1% insulin, 0.5% transferrin, 0.007% hydrocortisone, 0.1% fibronectin, 0.02% penicillin and streptomycin each), and were seeded into 12-well cell culture plates at a density of 5 × 10^5^ cells/mL to form a cell monolayer. The plates were incubated for 48 h at 37 °C under 5% CO_2_.

### 3.3. Determination of Lactobacillus Strains Tolerance to Gastric and Intestinal Stresses

Assays were performed as previously described in [59] with modifications.

#### 3.3.1. Gastric Stress Imitation In Vitro

A culture (100 µL) in the stationary phase grown in the MRC was diluted with 1 mL of the artificial gastric juice (dilution 1/11). The control was a culture (100 µL) in the stationary phase grown in the MRC that was diluted with 1 mL of MRS (dilution 1/11). 

The cultures were incubated for various periods of time (10 min, 30 min, and 60 min) at 37 °C in 10% CO_2_. Serial dilutions were plated onto MRS agar for estimation of colony-forming units (CFU/mL) counts. The composition of the artificial gastric juice was as follows: NaCl (Sigma S9625)—2.2 g/L; L-lactic acid (Sigma L1750)—9.9 g/L (0.11 M); and pepsin (porcine) (Sigma P7125) (600–1800 units/mg)—3.5 g/L; pH: 2.70 ± 0.02 (increased with 35% NaOH solution). The pH after a 1/11 dilution was 3.1 ± 0.1 (pH was controlled for each culture).

#### 3.3.2. Intestinal Stress Imitation In Vitro

A culture (100 µL) in the stationary phase grown in the MRC was diluted with 1 mL of the artificial intestinal juice 1mL (pH: 6.3). The cultures were incubated for 5 h at 37 °C in 10% CO_2_. The control was a culture (100 µL) in the stationary phase grown in the MRC that was diluted with 1 mL of MRS (dilution 1/11, incubation for 5 h at 37 °C in 10% CO_2_). Serial dilutions were plated onto MRS agar for estimation of colony-forming units (CFU/mL) counts. The composition of the artificial intestinal juice was as follows: bile salts (porcine bile, Sigma S8875)—3.3 g/L (final concentration 0.3%); and carbonate buffer NaHCO_3_ (Sigma S8875)—16.5 g/L (final concentration 1.5%). The final pH was 6.3.

The counting of microorganisms in a milliliter of culture (colony-forming units/mL—CFU/mL) was carried out according to the formula:$$\frac{\sum C}{\left(n_{1}+0.1\;n_{2}\right)d}\;,$$
where Σ*C*—the sum of all characteristic colonies counted on all Petri dishes containing from 15 to 300 colonies, *n*_1_—the number of cups in the lowest dilution (2 cups per dilution), *n*_2_—number of cups in the highest dilution (2 cups per dilution), and *d*—the value of the first dilution (low dilution) taken for counting.

The degree of resistance to gastric or intestinal stress RD (Resistance Degree) was determined by the formula: $$\mathrm{RD}=\frac{n_{1}}{n_{2}}\;,$$
where *n*_1_—the number of colony-forming cells per ml (CFU/mL) in the control and *n*_2_—the number of colony-forming cells per ml (CFU/mL) in the experiment.

The level of discrimination of gastric and intestinal stress was evaluated as “very good” when the RD (discrimination ratio) is less than 5, “good” when the RD is between 5 and 10, “acceptable” when the RD is between 10 and 15, and “unacceptable” when the RD exceeds 15.

### 3.4. Screening of Anti-Salmonella Lactobacilli by Delayed Antagonism Method

The antagonistic activity of lactobacillus strains LS7247, LA7234, and LG7528 was measured in vitro using the delayed antagonism method. A 2 × 10^9^ microbial cells/mL suspension of the studied Lactobacillus strain was sown with a stroke along the diameter of the Petri dish with a loop with a diameter of (3.5 ± 0.5) mm on an agarized medium. After 72 h of incubation at a temperature of 37 ± 1 °C in anaerobic conditions, cultures of SE or ST strains were sown perpendicular to the grown culture. A suspension of an SE or ST culture of 2–3 passages was prepared in 0.9% sodium chloride solution at a concentration of 5 × 10^8^ microbial cells/mL. The sowing of SE or ST strains was carried out with a 2 mm wide loop in the direction perpendicular to the growth zone of the lactobacillus strain. The cups were covered with a lid, placed in a thermostat, and incubated for 24 h at a temperature of 37 ± 1 °C. Preliminary assessment of the results was carried out after 24 h and 48 h, and the size of the growth inhibition zone of the *Salmonella* strains in mm was finally recorded. The greater the inhibition of the growth of the *Salmonella* strain, the higher the antagonistic activity of the lactobacillus strain.

### 3.5. Preparation of Cell-Free Supernatant (CFS) and ∆CFS from LS7247 Strain

Native cell-free supernatant (CFS) was prepared from cultures of the LS7247 strain as previously described in [113] with modifications. Briefly, the LS7247 strain was grown for 18 h in MRS broth under anaerobic conditions at 37 °C. The culture was diluted to a concentration of 1 × 10^8^ CFU/mL in MRS broth and further grown anaerobically for 48 h. CFS was collected by centrifugation at 6000× *g* for 25 min at 4 °C, filter-sterilized using a 0.22 µm pore size filter (Millipore, Bedford, MA, USA), and concentrated by speed-vacuum drying (Rotational Vacuum Concentrator RVC2-18, Martin Christ, Osterode am Harz, Germany). To obtain the ∆CFS, proteinase K (1.0 mg/mL) was added to native CFS from LS7247 and incubated at 37 °C for 2 h. 

Native CFS and ∆CFS from the LS7247 strain alone or together with the Actigen prebiotic (Alltech Inc., Nicholasville, KY, USA) were used to study the effectiveness of inhibiting the adhesion of SE and ST pathogens to human and animal enterocytes.

### 3.6. Determination of Anti-Salmonella Activity of CFS of One of LS7247 Strain

The CFS of LS7247 was additionally tested for anti-*Salmonella* activity according to [114] with modifications. LS7247 cells were grown overnight in MRS broth and centrifuged at 12,000 rpm for 20 min at 4 °C. The CFS was filter-sterilized using a 0.22 µm pore size filter (Millipore, USA). The CFS from the LS7247 strain was divided into 2 parts. The first part of CFS in a volume of 100 µL was placed into wells (6 mm) drilled into Muller Hinton agar, pre-seeded with approximately 10^5^ CFU/mL of the SE or ST strain. The second part of the CFS was titrated with NaOH to pH 7.0 and then 100 µL was placed in wells (6 mm) drilled in Muller Hinton agar, pre-seeded with about 10^5^ CFU/mL of the SE or ST strain. All agar plates were incubated at 37 °C for 10 h, and growth inhibition zones of SE or ST in mm were then measured.

### 3.7. Lactic Acid Determination in CFS of LS7247

The suspension of LS7247 cells grown in the MRS broth was centrifuged at 10,000× *g* for 15 min at 4 °C. The culture supernatant was filtered using a 0.22 µm pore size filter (Millipore). The concentration of lactic acid in the CFS produced by the LF3872 strain was determined according to the method in [115] using a high-performance liquid chromatography (HPLC) system equipped with an ultraviolet–visible detector (Shidmadzu, Kyoto, Japan) set at 220 nm and a Luna C18(2) column (150 mm × 4.6 mm, 5 µm; Phenomenex, Torrance, CA, USA). HPLC-grade lactic acid (Sigma-Aldrich, St. Louis, MO, USA) was used as the standard.

### 3.8. Identification of Genes Encoding Bacteriocins Produced by LS7247 Strain and Determination of Bacteriocin Primary Structure

Bacteriocin-related genes were identified using two methods: browsing GenBank annotations and utilizing the BAGEL4 program online (http://bagel4.molgenrug.nl/, accessed on 15 August 2023). These approaches allowed for the identification of putative bacteriocin-encoding genes from genome sequencing data [116]. The predicted primary structures of the bacteriocins produced by the LS7247 strain were determined using the nucleotide sequence of their structural genes.

### 3.9. Determination of Anti-Salmonella Activity of LS7247 Strain by Co-Cultivation Method in a Liquid Medium

The antibacterial activity of the LS7247 strain against the SE and ST strains was determined by co-cultivation in TGVC medium at a temperature of 37 ± 1 °C for 8 h according to [117] with modifications. Briefly, for co-cultivation, 1 mL of LS7247 inoculum grown on MRC medium (10^7^ CFU/mL) and the test SE or ST strains (10^7^ CFU/mL) were introduced into 20 mL of TGVC medium. In the preliminary experiments, it was found that all SE and ST strains and LS7247 strain grew on TGVC medium. The counting of SE and ST cells grown on TGVC medium in monoculture (control) and grown in the presence of lactobacilli on TGVC medium was carried out after 24 h. Aliquots of 1 mL were taken aseptically at 24 h, serially diluted, and spread onto selective xylose lysine deoxycholate (XLD) agar plates. All the plates were incubated for 24 h at 37 °C under aerobic conditions; at the end of incubation, the colonies of all SE and ST strains were counted and expressed as colony-forming units per milliliter (CFU/mL).

### 3.10. Assessment of Cytoplasmic Membrane Permeability of SE and ST Pathogens by Measurement of Extracellular ATP 

SE and ST pathogens at the logarithmic phase were centrifuged and resuspended in PBS (pH 7.0) with OD600 = 1.0. The pathogens were treated for 2.5 h at 37 °C in the presence of the CFS of LS7247 (100 µg/mL) and extracellular ATP levels after CFS treatment were detected by an ATP detection kit (Beyotime, Shanghai, China). Detection of luminescence was performed using an Infinite 200 PRO microplate reader (Tecan, Männedorf, Switzerland).

### 3.11. Determination of LS7247 Adhesion to a Monolayer Formed from Human Caco-2, Porcine IPEC-J2, or Chicken Primary Cecal Enterocytes 

Immortalized epithelial Caco-2 cells were used as a human in vitro intestinal epithelial model. Immortalized epithelial IPEC-J2 cells were used as a porcine in vitro intestinal epithelial model. Primary cecal epithelial cells were used as a chicken in vitro intestinal epithelial model. Caco-2 cells and IPEC-J2 cells were seeded into 12-well cell culture plates at a density of 5 × 10^5^ cells/mL and were grown to monolayers of immature cells with a layer of 80–100% confluency in Dulbecco’s modified Eagle’s medium (DMEM, Invitrogen, Waltham, MA, USA) supplemented with 20% heat-inactivated fetal bovine serum (FBS), 2 mM L-glutamine, penicillin (100 U/mL), and streptomycin (100 mg/mL). Chicken primary cecal epithelial cells were seeded into 12-well cell culture plates at a density of 5 × 10^5^ cells/mL and were grown in culture medium (DMEM plus 2.5% fetal calf serum, 0.1% insulin, 0.5% transferrin, 0.007% hydrocortisone, 0.1% fibronectin, and 0.02% penicillin and streptomycin each). 

The plates were incubated for 48 h at 37 °C under 5% CO_2_. The medium was replaced daily. LS7247 cells at a concentration of 10^9^ CFU/mL were introduced into wells containing monolayers of intestinal cells in a fresh nutrient medium. The plates were incubated at a temperature of +37 °C at 5% CO_2_ for 2 h and washed three times with sterile PBS to remove non-adherent bacteria, stained with azure–eosin, and examined under a Leica DM 4500B microscope (Leica, Calgary, AB, Canada). Adherent bacteria were quantified using the Leica IM modular applications system (Leica). The adhesion activity and the adhesion index were determined. The adhesion activity of the LS7247 strain is the percentage of enterocytes on the surface of which LS7247 cells are found. The adhesion index is the number of LS7247 cells adhered to the surface of one enterocyte.

### 3.12. Determination of Total Effects of Actigen Prebiotic and CFS from LS7247 Strain in Inhibiting the Adhesion of SE and ST Pathogens to Human Caco-2, Porcine IPEC-J2, and Chicken Primary Cecal Enterocytes 

Monolayers of Caco-2, IPEC-J2, and chicken primary cecal enterocytes were incubated in the presence of the Actigen prebiotic at a concentration of 40 µg/mL, CFS at concentration of 40 µg/mL, or mixture of Actigen at a concentration of 20 µg/mL + CFS at a concentration of 20 µg/mL in PBS; the control group was treated with only PBS for 1 h at 37 °C under 5% CO_2_. Then, the suspension of each SE or ST pathogen (5 × 10^7^ CFU/mL) was added to the monolayers of enterocytes. The plates were incubated for 2 h at 37 °C under 5% CO_2_. The monolayers of enterocytes were washed three times with sterile PBS to remove unbound SE or ST pathogens and CFS, Actigen, or the mixture, fixed with methanol, stained with azure–eosin, and examined under a Leica DM 4500B microscope (Leica). Adherent SE or ST pathogens were quantified using the Leica IM modular applications system (Leica). The adhesion of SE or ST pathogens to epithelial cells was expressed as the average number of adhering bacteria per epithelial cell.

### 3.13. Statistical Analysis

The results were analyzed using one-way analysis of variance (ANOVA) and represented as mean ± SD of six independent experiments, tested in triplicate. Statistical significance was evaluated by Student’s *t*-tests. The results were considered significant at *p* < 0.05.

## 4. Conclusions

The results of this study indicate that LS7247 has a promising potential as a probiotic. It exhibited high tolerance to both gastric and intestinal stresses. LS7247 effectively adhered to monolayers formed by human Caco-2, porcine IPEC-J2, and chicken primary cecal enterocytes, which is necessary for the long-term realization of its probiotic properties. LS7247 and its CFS demonstrated anti-*Salmonella* activity against SE and ST pathogens. Co-cultivation of LS7247 with SE and ST pathogens, including antibiotic-resistant strains, for 8 h resulted in a 1.0–1.5 log reduction in pathogen CFUs. Cell wall damage and ATP leakage in SE and ST pathogens were induced by CFS of LS7247. 

A cluster of genes located in two plasmids of LS7247 are responsible for the production of the bacteriocin belonging to class IIb and a bacteriocin called nisin. These bacteriocins, together with lactic acid produced by LS7247, increased the permeability of the outer membrane of SE and ST pathogens and created conditions for the penetration of the lytic complex Enterolysin A and metalloendopeptidase into these pathogens. A cluster of genes located in the chromosome of LS7247 are responsible for the production of Enterolysin A and metalloendopeptidase. The CFS of LS7247 in combination with the Actigen prebiotic demonstrated a significant combined effect in inhibiting the adhesion of SE and ST pathogens to human and animal enterocytes. The molecular mechanisms of these properties of the CFS have not been studied. 

Our future research will be aimed at elucidating the molecular mechanisms of the anti-adhesive properties of the CFS from LS7247 and its significant combined effect with the Actigen prebiotic in inhibiting the adhesion of SE and ST pathogens to human and animal enterocytes. The results obtained are of great importance for the development of a synbiotic formulation based on the LS7247 strain and Actigen prebiotic, or a synbiotic supplement utilizing them for the effective prevention of salmonellosis.

## Figures and Tables

**Figure 1 antibiotics-12-01535-f001:**
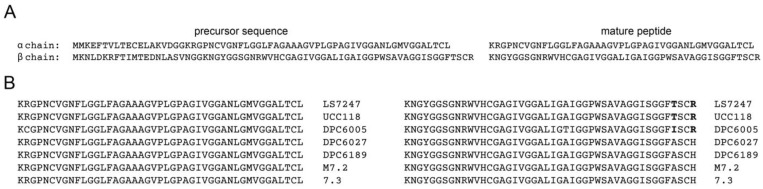
(**A**) Predicted amino acid sequences of class IIb bacteriocin IIbB LS7247 produced by LS7247 strain. (**B**) Multiple amino acid sequence alignments of alpha and beta peptide chains of class IIb bacteriocins produced by *L. salivarius* strains: LS7247, UCC118, DPC6005, DPC6027, DPC6189, M7.2, and 7.3. Amino acid residue differences are highlighted in bold.

**Figure 2 antibiotics-12-01535-f002:**
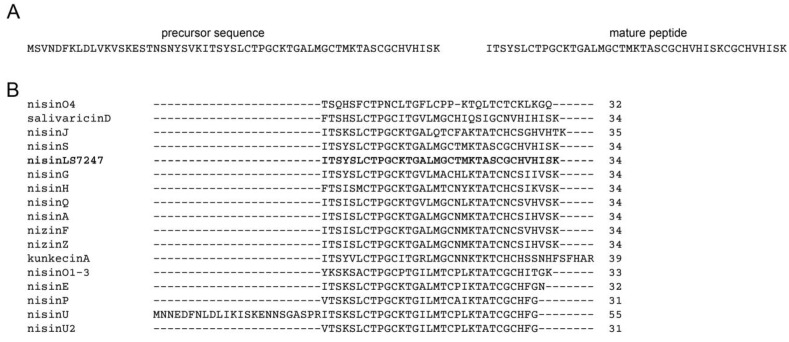
(**A**) The predicted primary structure of nisin (lantibiotic) based on the nucleotide sequence of the gene found in the genome of the LS7247 strain plasmid. (**B**) Alignment of primary structures of predicted nisins produced by LS7247 strain (highlighted in bold) with different nisins.

**Table 1 antibiotics-12-01535-t001:** Tolerance of lactobacillus strains to gastric and intestinal stresses in vitro.

N	Strain	Gastric Stress *						Intestinal Stress *	
		10 min		30 min		60 min		5 h	
		CFU/mL		CFU/mL		CFU/mL		CFU/mL	
		Experiment	Control	Experiment	Control	Experiment	Control	Experiment	Control
1	*L. animalis* IIE 7234	(2.40 ± 0.50) × 10^7^	(2.00 ± 0.60) × 10^8^	(1.41 ± 0.48) × 10^7^	(1.55 ± 0.63) × 10^8^	(1.35 ± 0.44) × 10^7^	(1.71 ± 0.51) × 10^8^	(5.12 ± 0.31) × 10^5^	(3.05 ± 0.47) × 10^8^
		RD = 8.1 ± 1.6 Good		RD = 10.7 ± 0.6 Acceptable		RD = 12.6 ± 0.9 Acceptable		RD = 610.0 ± 11.5 Unacceptable	
2	*L. salivarius* IIE 7247	(2.18 ± 0.61) × 10^7^	(2.35 ± 0.62) × 10^7^	(2.14 ± 0.59) × 10^7^	(2.41 ± 0.55) × 10^7^	(1.19 ± 0.44) × 10^7^	(2.36 ± 0.46) × 10^7^	(5.43 ± 0.62) × 10^7^	(2.00 ± 0.51) × 10^8^
		RD = 1.1 ± 0.1 Very good		RD = 1.1 ± 0.1 Very good		RD = 2.1 ± 0.8 Very good		RD = 4 ± 0.3 Very good	
3	*L. gasseri* IIE 7528	(8.82 ± 0.53) × 10^7^	(5.44 ± 0.62) × 10^8^	(2.19 ± 0.57) × 10^6^	(1.71 ± 0.65) × 10^7^	(1.20 ± 0.40) × 10^5^	(9.43 ± 0.48) × 10^5^	(1.05 ± 0.44) × 10^7^	(7.12 ± 0.82) × 10^8^
		RD = 6.1 ± 0.8 Good		RD = 7.6 ± 1.2 Good		RD = 7.8 ± 1.1 Good		RD = 67.8 ± 3.2 Unacceptable	

* Data are represented as means ± SD of three independent experiments, tested in triplicate.

**Table 2 antibiotics-12-01535-t002:** Anti-*Salmonella* activity of lactobacillus strains.

*Salmonella* Strain	*Salmonella* Strain Growth Inhibition Zone (nm)		
	*L. animalis* IIE 723	*L. salivarius* IIE 7247	*L. gasseri* IIE 7528
*S.* Enteritidis ATCC 13076	10.8 ± 0.4	16.5 ± 0.7 *	12.7 ± 0.5
*S.* Enteritidis ATCC 4931	11.2 ± 0.5	18.3 ± 0.6 *	10.9 ± 0.5
*S.* Enteritidis IIE Egg 6215	10.9 ± 0.5	17.4 ± 0.8 *	8.2 ± 0.5
*S.* Enteritidis IIE Egg 6218	8.7 ± 0.3	18.5 ± 0.4 *	9.5 ± 0.3
*S.* Enteritidis IIE Egg 6219	9.6 ± 0.6	18.9 ± 0.6 *	10.2 ± 0.4
*S.* Typhimurium ATCC 700720	8.4 ± 0.5	16.8 ± 0.5 *	11.6 ± 0.7
*S.* Typhimurium ATCC 14028	12.5 ± 0.6	17.6 ± 0.5 *	7.8 ± 0.6
*S.* Typhimurium IIE BR 6458	9.8 ± 0.4	16.9 ± 0.8 *	10.3 ± 0.5
*S.* Typhimurium IIE BR 6461	8.5 ± 0.3	18.6 ± 0.4 *	9.7 ± 0.3

Data are represented as means ± SD of six independent experiments, tested in triplicate. Asterisks indicate a significant difference in the anti-*Salmonella* activity of the *L. salivarius* IIE 7247 strain compared to the *L. animalis* IIE 7234 strain and *L. gasseri* IIE 7528 strain (* *p* < 0.05).

**Table 3 antibiotics-12-01535-t003:** Anti-*Salmonella* activity of CFS of LS7247 strain.

*Salmonella* Strain	*Salmonella* Strain Growth Inhibition Zone (mm)	
	CFS Intact	CFS Neutralized by NaOH
*S.* Enteritidis ATCC 13076	15.1 ± 0.6 *	6.5 ± 0.3
*S.* Enteritidis ATCC 4931	14.2 ± 0.4 *	5.4 ± 0.4
*S.* Enteritidis IIE Egg 6215	18.5 ± 0.6 *	5.8 ± 0.3
*S.* Enteritidis IIE Egg 6218	14.9 ± 0.7 *	6.9 ± 0.5
*S.* Enteritidis IIE Egg 6219	15.3 ± 0.5 *	6.2 ± 0.5
*S.* Typhimurium ATCC 700720	14.7 ± 0.5 *	6.8 ± 0.4
*S.* Typhimurium ATCC 14028	19.4 ± 0.6 *	5.7 ± 0.3
*S.* Typhimurium IIE BR 6458	14.5 ± 0.5 *	6.2 ± 0.4
*S.* Typhimurium IIE BR 6461	14.8 ± 0.4 *	5.9 ± 0.5

Data are represented as means ± SD of six independent experiments, tested in triplicate. * *p* < 0.05 *Salmonella* strain growth inhibition zone induced by CFS vs. CFS neutralized by NaOH.

**Table 4 antibiotics-12-01535-t004:** Dynamics of lactic acid production in the process of LS7247 strain cultivation in MRS broth.

Cultivation Time, h	4	24	48	72
Lactic acid production, mM	3.5 ± 0.4	69.7 ± 0.8 *	124.0 ± 2.5 **	41.8 ± 0.6 *

Data are represented as means ± SD of six independent experiments, tested in triplicate. * *p* < 0.01—lactic acid level in culture medium after 24 h of LS7247 strain cultivation vs. 4 h of cultivation; ** *p* < 0.001—lactic acid level in culture medium after 48 h of LS7247 strain cultivation vs. 4 h of cultivation.

**Table 5 antibiotics-12-01535-t005:** Anti-*Salmonella* activity of LS7247 strain.

*Salmonella* Strain	0 h		24 h	
	C ^1^	JC ^2^	C ^1^	JC ^2^
*S.* Enteritidis ATCC 13076	2 × 10^5^	3 × 10^5^	8 × 10^5^	8 × 10^4^
*S.* Enteritidis ATCC 4931	3 × 10^5^	3 × 10^5^	8 × 10^5^	4 × 10^4^
*S.* Enteritidis IIE Egg 6215	4 × 10^5^	4 × 10^5^	2 × 10^6^	5 × 10^4^
*S.* Enteritidis IIE Egg 6218	3 × 10^5^	3 × 10^5^	9 × 10^5^	3 × 10^4^
*S.* Enteritidis IIE Egg 6219	4 × 10^5^	4 × 10^5^	8 × 10^5^	2 × 10^4^
*S.* Typhimurium ATCC 700720	3 × 10^5^	3 × 10^5^	8 × 10^5^	2 × 10^4^
*S.* Typhimurium ATCC 14028	4 × 10^5^	4 × 10^5^	2× 10^6^	5 × 10^4^
*S.* Typhimurium IIE BR 6458	3 × 10^5^	3 × 10^5^	8 × 10^5^	4 × 10^4^
*S.* Typhimurium IIE BR 6461	4 × 10^5^	4 × 10^5^	2 × 10^6^	4 × 10^4^

^1^ Control; number of *Salmonella* cells in monoculture (CFU/mL). ^2^ Number of *Salmonella* cells in co-culture with LS7247 (CFU/mL). Data are representative of six independent experiments, tested in triplicate.

**Table 6 antibiotics-12-01535-t006:** Extracellular ATP levels in SE and ST pathogens treated with CFS of LS7247.

*Salmonella* Strain	Control ^1^	CFS LS7247 ^2^
*S.* Enteritidis ATCC 13076	5.7 ± 0.8	25.4 ± 1.2 *
*S.* Enteritidis ATCC 4931	4.6 ± 0.7	23.7 ± 1.0 *
*S.* Enteritidis IIE Egg 6215	5.3 ± 0.9	28.5 ± 1.3 *
*S.* Enteritidis IIE Egg 6218	4.5 ± 0.6	24.3 ± 1.2 *
*S.* Enteritidis IIE Egg 6219	5.9 ± 0.8	27.5 ± 1.2 *
*S.* Typhimurium ATCC 700720	5.4 ± 0.9	28.4 ± 1.1 *
*S.* Typhimurium ATCC 14028	4.8 ± 0.7	25.8 ± 1.0 *
*S.* Typhimurium ATCC 14028	5.6 ± 0.5	29.5 ± 1.2 *
*S.* Typhimurium IIE BR 6461	4.0 ± 0.5	23.9 ± 1.1 *

^1,2^ Concentration of ATP (nm/OD). Control ^1^: Suspension of *Salmonella* cells in MRC medium. CFS of LS7247 ^2^: Suspension of *Salmonella* cells in CFS of LS7247 strain. * *p* < 0.05 extracellular ATP level in suspension of *Salmonella* cells in MRC medium vs. suspension of *Salmonella* cells in CFS of LS7247 strain. All data are representative of six independent experiments, tested in triplicate.

**Table 7 antibiotics-12-01535-t007:** LS7247 strain indicators for its ability to adhere to a monolayer formed from human Caco-2, porcine IPEC-J2, or chicken primary cecal enterocytes.

Adhesion Indicator of LS7247 Strain	Human and Animal Enterocytes		
	Human Caco-2	Porcine IPEC-J2	Chicken Cecal Cells
Adhesion activity	100%	100%	100%
Adhesion index	38.6 ± 2.5	32.4 ± 1.9	27.5 ± 1.6

Data are presented as the means ± SD of six independent experiments, tested in triplicate.

**Table 8 antibiotics-12-01535-t008:** Total effects of the Actigen prebiotic and CFS from LS7247 strain in inhibiting the adhesion of SE and ST pathogens to human Caco-2 enterocytes.

*Salmonella* Strain	PBS (Control)	Actigen ^1^	CFS ^2^	MIXT ^3^	∆CFS ^4^	∆MIXT ^5^
*S.* Enteritidis ATCC 13076	25.5 ± 1.2	6.4 ± 0.8 **	9.8 ± 1.0 **	0.75 ± 0.04 ***	27.4 ± 1.3	6.9 ± 0.7 **
*S.* Enteritidis ATCC 4931	28.3 ± 1.5	6.7 ± 0.5 **	9.5 ± 1.2 **	0.69 ± 0.05 ***	28.6 ± 1.5	5.7 ± 0.9 **
*S.* Enteritidis IIE Egg 6215	28.7 ± 1.3	5.9 ± 0.6 **	8.4 ± 1.1 **	0.84 ± 0.06 ***	23.9 ± 1.1	6.1 ± 0.5 **
*S.* Enteritidis IIE Egg 6218	26.4 ± 1.2	6.3 ± 0.7 **	8.6 ± 1.2 **	0.65 ± 0.03 ***	29.5 ± 1.6	6.4 ± 0.8 **
*S.* Enteritidis IIE Egg 6219	27.3 ± 1.4	5.8 ± 0.5 **	9.3 ± 1.2 **	0.56 ± 0.03 ***	24.7 ± 1.1	5.9 ± 0.7 **
*S.* Typhimurium ATCC 700720	26.5 ± 1.2	5.5 ± 0.8 **	8.5 ± 1.1 **	0.59 ± 0.04 ***	28.2 ± 1.4	6.5 ± 0.8 **
*S.* Typhimurium ATCC 14028	28.5 ± 1.4	5.9 ± 0.7 **	9.2 ± 1.3 **	0.62 ± 0.03 ***	25.6 ± 1.8	6.3 ± 0.7 **
*S.* Typhimurium IIE BR 6458	27.6 ± 1.5	6.2 ± 0.5 **	8.7 ± 1.1 **	0.67 ± 0.04 ***	24.8 ± 1.5	6.5 ± 0.5 **
*S.* Typhimurium IIE BR 6461	29.2 ± 1.4	6.8 ± 0.4 **	9.5 ± 1.2 **	0.58 ± 0.03 ***	26.7 ± 1.4	6.8 ± 0.7 **

^1^—Actigen prebiotic (concentration: 40 µg/mL); ^2^—lyophilized CFS (concentration: 40 µg/mL); ^3^—Mixture of CFS (concentration: 20 µg/mL) and Actigen prebiotic (concentration: 20 µg/mL); ^4^—lyophilized ∆CFS after cultivation with proteinase K (concentration: 40 µg/mL); ^5^—mixture of Actigen prebiotic (concentration: 20 µg/mL) and lyophilized ∆CFS (concentration: 20 µg/mL); ** *p* < 0.01 adhesion of SE and ST pathogens to Caco-2 alone vs. adhesion of SE and ST pathogens to Caco-2 + Actigen or adhesion of SE and ST pathogens to Caco-2 + CFS; *** *p* < 0.001 adhesion of SE and ST pathogens to Caco-2 alone vs. adhesion of SE and ST pathogens to Caco-2 + Actigen + CFS. Data are presented as the means ± SD of six independent experiments, tested in triplicate.

**Table 9 antibiotics-12-01535-t009:** Total effects of the Actigen prebiotic and CFS from LS7247 strain in inhibiting the adhesion of SE and ST pathogens to porcine IPEC-J2 enterocytes.

*Salmonella* Strain	PBS (Control)	Actigen ^1^	CFS ^2^	MIXT ^3^	∆CFS ^4^	∆MIXT ^5^
*S.* Enteritidis ATCC 13076	28.6 ± 1.3	5.4 ± 0.4 **	8.7 ± 1.3 **	0.64 ± 0.05 ***	26.4 ± 1.3	6.3 ± 0.4 **
*S.* Enteritidis ATCC 4931	27.5 ± 1.2	6.2 ± 0.2 **	9.5 ± 1.4 **	0.55 ± 0.03 ***	28.5 ± 1.2	5.8 ± 0.5 **
*S.* Enteritidis IIE Egg 6215	29.4 ± 1.5	6.5 ± 0.3 **	8.3 ± 1.2 **	0.49 ± 0.05 ***	27.6 ± 1.4	6.1 ± 0.3 **
*S.* Enteritidis IIE Egg 6218	26.9 ± 1.4	5.2 ± 0.4 **	7.9 ± 1.1 **	0.46 ± 0.04 ***	29.5 ± 1.2	7.4 ± 0.8 **
*S.* Enteritidis IIE Egg 6219	25.4 ± 1.1	5.3 ± 0.4 **	8.2 ± 1.5 **	0.48 ± 0.05 ***	24.8 ± 1.6	6.5 ± 0.5 **
*S.* Typhimurium ATCC 700720	27.2 ± 1.4	7.2 ± 0.3 **	9.4 ± 1.3 **	0.59 ± 0.04 ***	28.9 ± 1.7	7.1 ± 0.4 **
*S.* Typhimurium ATCC 14028	25.8 ± 1.3	5.2 ± 0.6 **	8.2 ± 1.2 **	0.45 ± 0.03 ***	29.2 ± 1.5	7.2 ± 0.8 **
*S.* Typhimurium IIE BR 6458	26.7 ± 1.1	5.7 ± 0.4 **	8.4 ± 1.5 **	0.48 ± 0.05 ***	28.4 ± 1.4	6.9 ± 0.7 **
*S.* Typhimurium IIE BR 6461	28.2 ± 1.6	7.5 ± 0.9 **	9.3 ± 1.1 **	0.67 ± 0.04 ***	29.6 ± 1.7	7.1 ± 0.5 **

^1^—Actigen prebiotic (concentration: 40 µg/mL); ^2^—lyophilized CFS (concentration: 40 µg/mL); ^3^—mixture of Actigen prebiotic (concentration: 20 µg/mL) and lyophilized CFS (concentration: 20 µg/mL); ^4^—lyophilized ∆CFS after cultivation with proteinase K (concentration: 40 µg/mL); ^5^—mixture of Actigen prebiotic (concentration: 20 µg/mL) and lyophilized ∆CFS (concentration: 20 µg/mL); ** *p* < 0.01 adhesion of SE and ST pathogens to IPEC-J2 alone vs. adhesion of SE and ST pathogens to IPEC-J2 + Actigen or adhesion of SE and ST pathogens to IPEC-J2 + CFS; *** *p* < 0.001 adhesion of SE and ST pathogens to IPEC-J2 alone vs. adhesion of SE and ST pathogens to IPEC-J2 + Actigen + CFS. Data are presented as the means ± SD of six independent experiments, tested in triplicate.

**Table 10 antibiotics-12-01535-t010:** Total effects of the Actigen prebiotic and CFS from LS7247 strain in inhibiting the adhesion of SE and ST pathogens to chicken primary cecal enterocytes.

*Salmonella* Strain	PBS (Control)	Actigen ^1^	CFS ^2^	MIXT ^3^	∆CFS ^4^	∆MIXT ^5^
*S.* Enteritidis ATCC 13076	19.4 ± 1.5	4.8 ± 0.3 **	6.5 ± 0.8 **	0.5 ± 0.04 ***	23.5 ± 1.7	5.4 ± 0.5 **
*S.* Enteritidis ATCC 4931	21.7 ± 1.3	5.8 ± 0.6 **	6.7 ± 0.5 **	0.7 ± 0.05 ***	24.8 ± 1.3	4.9 ± 0.6 **
*S.* Enteritidis IIE Egg 6215	18.6 ± 1.6	4.9 ± 0.4 **	5.9 ± 0.7 **	0.6 ± 0.03 ***	19.7 ± 1.1	5.7 ± 0.4 **
*S.* Enteritidis IIE Egg 6218	22.3 ± 1.4	4.6 ± 0.3 **	6.3 ± 0.6 **	0.4 ± 0.02 ***	20.9 ± 1.6	5.1 ± 0.3 **
*S.* Enteritidis IIE Egg 6219	19.5 ± 1.2	4.7 ± 0.5 **	5.7 ± 0.4 **	0.5 ± 0.03 ***	22.3 ± 1.8	4.8 ± 0.5 **
*S.* Typhimurium ATCC 700720	23.6 ± 1.1	5.3 ± 0.4 **	6.6 ± 0.7 **	0.6 ± 0.04 ***	19.9 ± 1.2	5.3 ± 0.4 **
*S.* Typhimurium ATCC 14028	20.4 ± 1.3	5.7 ± 0.5 **	7.5 ± 0.4 **	0.6 ± 0.03 ***	21.4 ± 1.5	5.2 ± 0.6 **
*S.* Typhimurium IIE BR 6458	21.7 ± 1.5	5.2 ± 0.4 **	7.2 ± 0.8 **	0.8 ± 0.05 ***	22.7 ± 1.8	4.9 ± 0.5 **
*S.* Typhimurium IIE BR 6461	22.5 ± 1.6	4.9 ± 0.3 **	6.8 ± 0.5 **	0.7 ± 0.04 ***	20.6 ± 1.3	4.7 ± 0.6 **

^1^—Actigen prebiotic (concentration: 40 µg/mL); ^2^—lyophilized CFS (concentration: 40 µg/mL); ^3^—mixture of Actigen prebiotic (concentration: 20 µg/mL) and lyophilized CFS (concentration: 20 µg/mL); ^4^—lyophilized ∆CFS after cultivation with proteinase K (concentration: 40 µg/mL); ^5^—mixture of Actigen prebiotic (concentration: 20 µg/mL) and lyophilized ∆CFS (concentration: 20 µg/mL); ** *p* < 0.01 adhesion of SE and ST pathogens to CPCE alone vs. adhesion of SE and ST pathogens to CPCE + Actigen or adhesion of SE and ST pathogens to CPCE + CFS; *** *p* < 0.001 adhesion of SE and ST pathogens to CPCE alone vs. adhesion of SE and ST pathogens to CPCE + Actigen + CFS. Data are presented as the means ± SD of six independent experiments, tested in triplicate.

**Table 11 antibiotics-12-01535-t011:** Microorganisms used in this study.

Microorganism	Strain	Antibiotic Resistance	Growth Conditions
*L. salivarius*	IIE ^1^ LS7247 ^2^		MRS ^a^ 37 °C in CO_2_ incubator, 10% CO_2_ or anaerobically 48 h
*L. animalis*	IIE LA 7234 ^3^		The same
*L. gasseri*	IIE LG 7528 ^4^		The same
*S.* Enteritidis	ATCC 13076		BHI ^b^ 37 °C aerobically 18 h
*S.* Enteritidis	ATCC 4931		The same
*S.* Enteritidis	IIE Egg 6215 ^5^	NAL/AMP	The same
*S.* Enteritidis	IIE Egg 6218	AMP/TET/CIP/NAL/CHL	The same
*S.* Enteritidis	IIE Egg 6219	AMP/TET/CIP/NAL/AZM	The same
*S.* Typhimurium	ATCC 700720		The same
*S.* Typhimurium	ATCC 14028		The same
*S.* Typhimurium	IIE Br 6458 ^6^	NAL/AMP/TET	The same
*S.* Typhimurium	IIE Br 6461	AMP/TET/SXT/AZM	The same

^1^ Collection of Microorganisms at the Institute of Immunological Engineering (IIE), Department of Biochemistry of Immunity and Biodefence, Lyubuchany, Moscow Region, Russia. ^2^ Isolate from the intestines (the analysis of feces was carried out) and reproductive system (the analysis of the vaginal discharge was carried out) of a healthy woman. ^3^ Isolate from a broiler chicken intestine. ^4^ Isolate from a piglet intestine. ^5^ Isolate from chicken’s eggs. ^6^ Isolate from fresh broiler chicken meat. ^a^ Man–Rogosa–Sharpe (MRS) broth or agar-containing plates (HiMedia, India). ^b^ Brain-Heart Infusion (BHI) broth supplemented with 0.5% yeast extract or agar containing BHI plates. Antibiotic resistance: NAL—nalidixic acid, AMP—ampicillin, TET—tetracycline, CIP—ciprofloxacin, SXT—tri methoprimsulfamethoxazole, CHL—chloramphenicol, AZM—azithromycin.

## Data Availability

Not applicable.

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
