# Peer review of "Ligilactobacillus salivarius 7247 Strain: Probiotic Properties and Anti-Salmonella Effect with Prebiotics"

_antibiotics, 2023, doi:10.3390/antibiotics12101535_

Round 1

Reviewer 1 Report

REVIEW

Dear authors,

The work is focused on proposing the Ligilactobacillus salivarius 7247 strain as a potential probiotic with anti-Salmonella activity, for which I consider that the strategy used from the description of the strain at the genomic level, underlies much of the in vitro research carried out, as well as the demonstration of the role played by genetic elements in this LS7247 strain such as plasmids in the mechanisms against the different strains of Salmonella evaluated. The use of the prebiotic favors the microbicidal mechanisms of the LS7247 strain.

It is necessary to make the following corrections in the indicated lines:

Line 38: write the complete abbreviation of “LS7247”.

Lines 50, 769: write the term correctly “metalloendopeptidase”.

Line 420: correct strain name Salmonella “Enteritidis”.

Table 3: correct strain name Salmonella “Enteritidis ATCC 13076”.

Lines 746, 759: write the name in cursive “Salmonella”. 

Please amend the requested comments and submit the revision file.

Reviewer 2 Report

Abstract must be shortened according to the authors instruction - more briefly, mainly significant results

Line 210 - "Primary epithelial cells were obtained from the cecal of 2-week-old chicks" - I would like to know how you achieved the isolation of the sterile cells necessary for the establishment of the primary cell culture, because usually are contaminated

Reviewer 3 Report

 The manuscript has a good research theme and well written with the findings. However, there are few corrections that has be to done before publishing. Below are my comments, 

Line 39:  S. Enteritidis (SE) and S. Typhimurium (ST),  should be in italics.

 Line 44:  LA7234 and LG7528 strains, were these commercially available strains used as control for this experiment? Or isolated from any other source?

 Line 48:  It has been shown that LS7247 47 produced high level of LA. Does it mean after 48 hrs of cultivation or at the end of fermentation? If it’s produced high after 48 hrs of cultivation, pls continue it with the following sentence. If not, explain properly.

 Line 69: Multi Drug Resistant (MDR). This is the first time mentioning MDR in this article. So, abbreviate it

 In Table 1, Please write the bacterial names in italics

 Lines 266-268: Write the sentences using past tense

 Section, 3.4, lines 462-464, Mention how the lactic acid was measured from time to time?

General comments:

 -   Few sections were lengthy and revise, remove the sentences if think not       necessary.

Please keep the bacterial names in italics throughout the manuscript.

- Some sentences were not written in the past tense. please keep the uniform format throughout.

 The results and discussion section is well-written.

 Good luck with submission.

The manuscript reads well. It can be considered for publishing after making the above suggested corrections. 
